# Effects of Smartphone Size and Hand Size on Grip Posture in One-Handed Hard Key Operations

**Younggeun Choi [1], Xiaopeng Yang [2], Jangwoon Park [3] , Wonsup Lee [4] and Heecheon You [1,*]**

[1] Department of Industrial and Management Engineering, Pohang University of Science and Technology, Pohang 37673, Korea; sidek@postech.ac.kr

[2] School of Artificial Intelligence and Computer Science, Jiangnan University, Wuxi 214122, China; yxp233@jiangnan.edu.cn

[3] Department of Engineering, Texas A&M University–Corpus Christi, Corpus Christi, TX 78412, USA; jangwoon.park@tamucc.edu

[4] School of Global Entrepreneurship and Information Communication Technology, Handong Global University, Pohang 37554, Korea; w.lee@handong.edu

\* Correspondence: hcyou@postech.ac.kr



**Featured Application: The grip posture information identified from this study can provide fundamental knowledge of how users grasp a smartphone and be of use for the design of smartphone user interface.**

**Abstract:** Greater understanding of the grip postures preferred by users is needed for the ergonomic design of smartphone user interfaces. The present study identifies user-preferred grip postures by smartphone size and hand size in one-handed hard key operations. Grip postures of 45 participants were photographed while major smartphone tasks were simulated in standing with smartphone mockups of nine screen sizes (3.0″–7.0″). The grip postures were encoded by the locations (left side: L, right side: R, top: T, bottom: B, front: F, back: K) of a smartphone and the number of fingers at each contact location. Three grip postures (L3-R1-K1: 70.0%, L4-R1: 13.3%, L3-R1-T1: 12.0%) were found dominant and the distribution of grip posture changed by smartphone size and hand size—the larger the smartphone size or hand size, the higher the frequency of L3-R1-K1. The grip posture frequency distribution by smartphone size would be of use to determine the optimal locations of hard keys on a smartphone of a particular size.

**Keywords:** smartphone; hard key location; grip posture; posture encoding; frequency analysis

## 1. Introduction

Greater understanding user-preferred grip postures in one-handed operation of hard keys on a smartphone is needed to determine their optimal locations for better operational efficiency and comfort. Hard keys are used on a smartphone for volume adjustment, power on/off, and screen on/off. The operational efficiency of the hard keys can be improved if they are properly located based on the information of user-preferred grip postures. Improperly designed locations of the hard keys may lead to discomfort and inefficiency in finger motions due to awkward hand posture [1,2]. Since one-handed (single hand) hard key operations have more usability problems than two-handed (both hands) hard key operations in terms of mobility of the fingers, comfort, efficiency of key operation, and stability of grip [3–5], the locations of the hard keys need to be determined by considering user-preferred grip postures in a one-handed hard key operation.

No research has been conducted to examine the effects of task type, device size, and hand size on user-preferred grip posture to determine the proper locations of hard keys on a hand-held mobile

device. Various tasks such as checking an email, searching on the internet, taking a photo, sending a text message, making a call, and answering a call require different grip postures with different frequencies [6]. For example, in a one-handed operation, a light grip is used for searching on the internet by moving the thumb on the touch screen while supporting the back of the smartphone with the four fingers, while a firm grip is used for locking/unlocking the phone by grasping the sides of the phone firmly to apply a significant force to the power key. Next, Pelosi et al. analyzed the effects of finger location on antenna performance parameters in various grip postures for talk and data modes in mobile phones: soft grip (the sides of a phone are held by the distal phalanges) and firm grip (the sides of a phone are held by the intermediate phalanges) in one-handed operations [7]. Myllymaki et al. investigated various arrangements of capacitive proximity sensors on a mobile phone for recognition of grip position [8]. Wobbrock et al. compared four one-handed grip postures (thumb-on-front, thumb-on-back, index-on-front, and index-on-back) in terms of task completion time and selection error rate in a target area selection task on the touch screen of a personal digital assistant (PDA) [2]. Lastly, Tu et al. investigated user-defined gestures on a tablet PC depending on three holding postures [9,10]. However, the existing studies do not examine user-preferred grip postures for smartphone hard key operation tasks, which are important to determine the proper locations of hard keys on a smartphone for operation efficiency and comfort.

The present study was intended to identify grip postures preferred by users with various hand sizes in one-handed hard key operations for major smartphone tasks with smartphones of various screen sizes. Grip postures captured by web cameras were encoded by the locations of a smartphone and the number of fingers at each contact location and the frequencies of grip postures were analyzed. Lastly, the effects of smartphone size and hand size on user-preferred grip posture were analyzed.

## 2. Materials and Methods

### 2.1. Participants

Smartphone users with more than one year of experience using mainly the right hand for one-handed operations and having no history of visual impairments and musculoskeletal disorders in their hands participated in the grip posture experiment of the present study. Nine participant groups in combination of three hand length and three hand width categories were defined as shown in Figure 1 by considering the hand size distributions of the Korean male and female populations [11] to recruit a representative group of smartphone users in their 20s to 50s in terms of hand size. A total of 45 smartphone users (28 males and 17 females; age = 24.8 ± 4.7; hand length = 180.6 ± 10.7 mm; hand width = 80.8 ± 6.7 mm) who participated in the study were found to be not significantly different from the Korean population ($n$ = 3921; hand length = 178.4 ± 10.5 mm; hand width = 79.4 ± 6.0 mm) in terms of mean ($t_{44}$ = 1.37, $p$ = 0.177 for hand length; $t_{44}$ = 1.41, $p$ = 0.165 for hand width) and variability ($F_{44, 3920}$ = 1.03, $p$ = 0.842 for hand length; $F_{44, 3920}$ = 1.23, $p$ = 0.287 for hand width) of hand size by $t$-test and $F$-test.

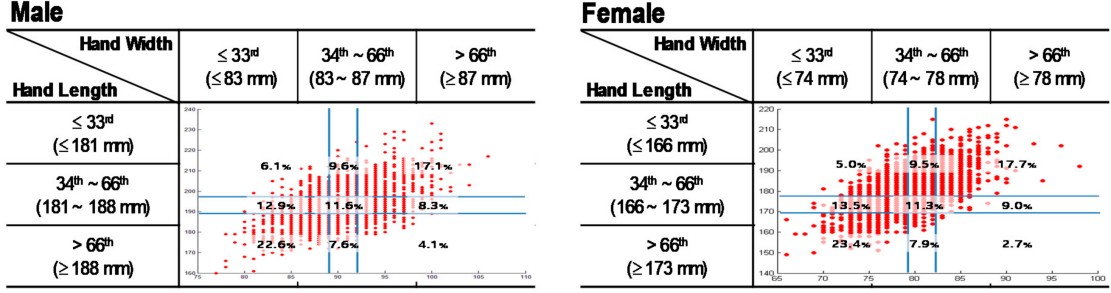

**Figure 1.** Hand size distribution of participants (28 males and 17 females) by gender.

### 2.2. Appratus

Nine smartphone mockups in different sizes and weights were prepared for the experiment. The dimensions of the smartphone mockups (screen size = 3.0–7.0 inches; height = 95–175 mm; width = 56–93 mm; depth = 8–12 mm; weight = 100–190 g) were determined as shown in Figure 2 based on a survey in the present study on the specifications of smartphones with significant market shares. The mockups were fabricated using the 3D printer Dimension SST 768 (Stratasys Ltd., Edina, MN, USA) to control the effects of affective factors such as brand, color, material, and finish on preferred grip posture. A sheet of paper screen was glued on each mockup to provide a graphical user interface when simulating smartphone tasks. Lastly, a lead sheet of a designated weight was attached to the inside of the mockup while keeping its center of mass in the middle.

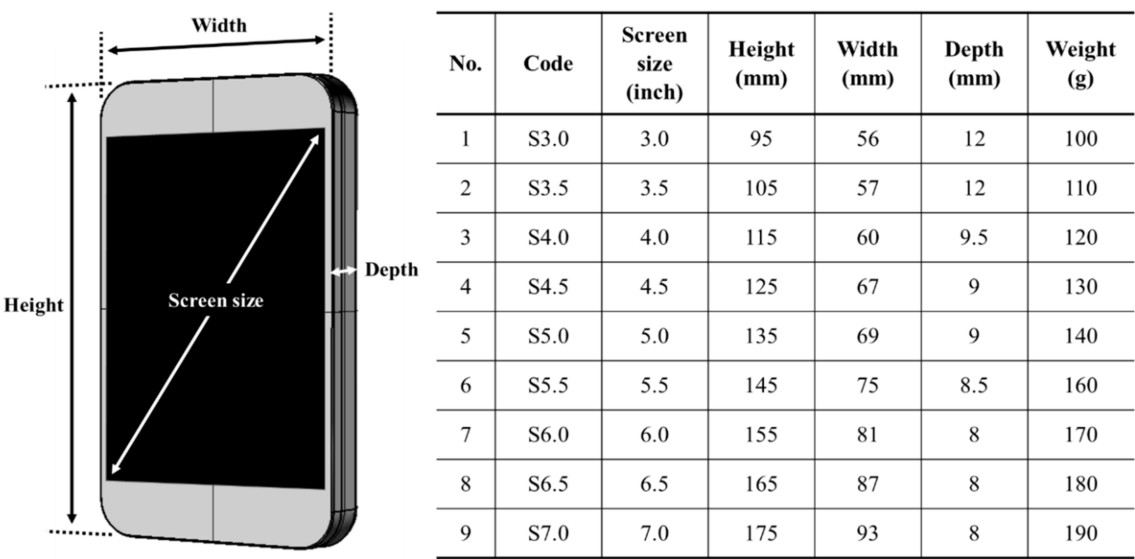

| No. | Code | Screen size (inch) | Height (mm) | Width (mm) | Depth (mm) | Weight (g) |
|---|---|---|---|---|---|---|
| 1 | S3.0 | 3.0 | 95 | 56 | 12 | 100 |
| 2 | S3.5 | 3.5 | 105 | 57 | 12 | 110 |
| 3 | S4.0 | 4.0 | 115 | 60 | 9.5 | 120 |
| 4 | S4.5 | 4.5 | 125 | 67 | 9 | 130 |
| 5 | S5.0 | 5.0 | 135 | 69 | 9 | 140 |
| 6 | S5.5 | 5.5 | 145 | 75 | 8.5 | 160 |
| 7 | S6.0 | 6.0 | 155 | 81 | 8 | 170 |
| 8 | S6.5 | 6.5 | 165 | 87 | 8 | 180 |
| 9 | S7.0 | 7.0 | 175 | 93 | 8 | 190 |

**Figure 2.** Diagram of a smartphone mockup and the specifications of nine smartphone mockups used in the grip posture experiment.

### 2.3. Experimental Procedure

The grip posture experiment was conducted in three phases: (1) introduction, (2) task familiarization, and (3) main experiment. In the introduction phase, the objectives and procedure of the experiment were explained to the participant and then written informed consent was obtained. In the task familiarization phase, the participant practiced the experiment to be familiarized with the mockups and operating the hard keys while conducting major smartphone tasks such as answering a call, listening to music, texting, and browsing the web, as shown in Table 1. The major smartphone tasks were selected by referring to previous studies such as [6,12–16] which investigated user performance of smartphones. A set of actions were specified for each major task so that one or two volume or power key operations could be conducted. For example, a texting task was performed in five actions: (1) turn on the screen by pressing the power key, (2) navigate screens by flicking for a message app, (3) select a message app by tapping, (4) enter and send a text, and (5) return to the home screen by pressing the home key. Lastly, in the main experiment phase, the participant simulated the tasks with a mockup in standing. The mockups in different sizes were tested in random order and the tasks with a mockup were performed in random order. Six grip postures (three for each of volume and power key operations) were captured for each of the mockups by two LifeCam Studio (Microsoft Co. Ltd., Redmond, WA, USA) web cameras. The participant was instructed to simulate a smartphone task with his/her comfortable grip and move the hand with the mockup in grip into the mid position between the cameras vertically arranged apart, as shown in Figure 3a, immediately after completing a hard key operation. Image frames of hard-key manipulation actions were extracted from the recorded video

images. Then, each image was manually encoded by indicating the locations (left side: L, right side: R, top: T, bottom: B, front: F, and back: K) of the mockup and the number of fingers at each contact location. For example, a grip posture captured for a power key operation in Figure 3b is coded as L3-R1-K1 (three fingers on the left side, one finger on the right side, and one finger on the back of the mockup). The present study was approved by the Institutional Review Board at Pohang University of Science and Technology.

**Table 1.** Major smartphone tasks and corresponding actions including operating hard keys (in italic).

| Tasks | Actions |
|---|---|
| Answering a call | 1. Grasp the phone;<br>2. Flick the screen to answer a call;<br>3. Place the phone to the ear to answer a call;<br>4. Adjust the volume up and down with *volume key*. |
| Listening to music | 1. Grasp the phone;<br>2. Adjust the volume up and down with *volume key*;<br>3. Navigate the screen;<br>4. Select a music app;<br>5. Check the list of songs;<br>6. Select a song;<br>7. Adjust the volume up and down with *volume key*. |
| Texting | 1. Turn on the screen with *power key*;<br>2. Navigate the screen;<br>3. Select a message app;<br>4. Send a message;<br>5. Return to home. |
| Web browsing | 1. Turn on the screen with *power key*;<br>2. Turn on Wi-Fi;<br>3. Select a web browser app;<br>4. Browse the internet;<br>5. Turn off the screen with *power key*. |

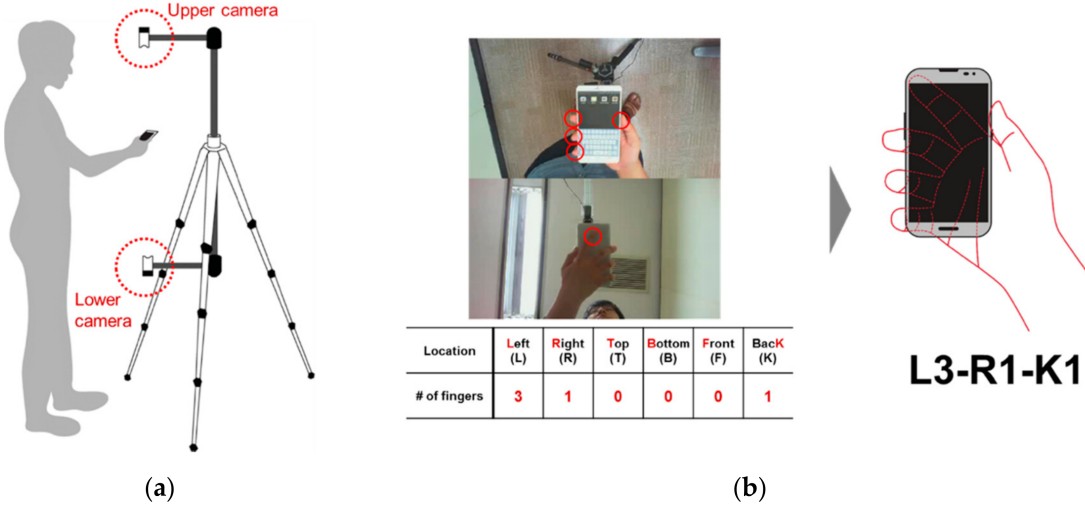

(**a**)            (**b**)

**Figure 3.** Measurement and encoding of smartphone grip posture (illustrated). (**a**) video recording of grip posture; (**b**) coding of grip posture.

## 2.4. Statistical Analysis

A chi-square test for independence was performed to identify the effects of smartphone size and hand size (hand length and hand width) on user-preferred grip posture for hard key operations by one

hand. The statistical testing was conducted using Minitab v. 16 (Minitab, Inc., State College, PA, USA) at $\alpha = 0.05$.

## 3. Results

A total of nine grip postures were identified from the experiment for power and volume key operations on the mockups, and of them, three grip postures (L3-R1-K1, L4-R1, and L3-R1-T1) were found to be dominant, having a total of 95% of use frequency, as shown in Figure 4. The grip posture L3-R1-K1 (holding the left side of a smartphone with three fingers, the right side with the thumb, and the back with the index finger) was most dominant with a frequency of 70.0% over 2430 (= 45 participants × 9 device sizes × 2 hard keys × 3 repetitions) grip posture measurements. Lastly, L4-R1 (holding the left side with the four fingers and the right side with the thumb) and L3-R1-T1 (holding the left side with three fingers, the right side with the thumb, and the top with the index finger) were the second and third most dominant grip postures with 13.3% and 12.0% of frequency for hard key operations, respectively.

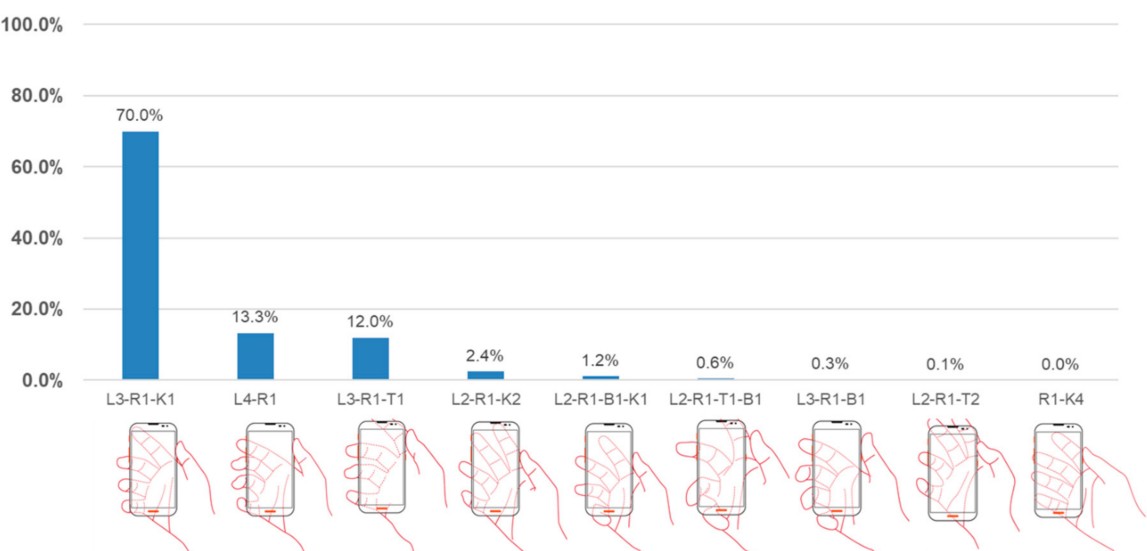

**Figure 4.** The use frequency distribution of grip posture for operation of the hard keys of a smartphone.

The frequency distribution of grip posture varied significantly by smartphone size, as shown in Figure 5 ($\chi^2(12) = 674.8$, $p < 0.001$). The frequency of L3-R1-K1 linearly increased from 32.2% to 82.6% over the smartphone size range of 3.0 to 5.0 inches and then levelled off in the smartphone size range of 5.0 to 7.0 inches, while that of L4-R1 gradually decreased from 21.1% to 6.7% over the smartphone size range of 3.0 to 7.0 inches and that of L3-R1-T1 linearly decreased from 39.3% to 2.2% over the smartphone size range of 3.0 to 5.0 inches and became negligible (<1%) in the smartphone size range of 5.5 to 7.0 inches. Lastly, the skewness of a frequency distribution of the three dominant grip postures increased toward L3-R1-K1 as smartphone size increased: a slightly skewed frequency distribution (32.2% for L3-R1-K1, 39.3% for L3-R1-T1, and 21.1% for L4-R1) for the 3.0-inch smartphone to a highly skewed frequency distribution (84.4% for L3-R1-K1, <0.1% for L3-R1-T1, and 6.7% for L4-R1) for the 7.0-inch smartphone.

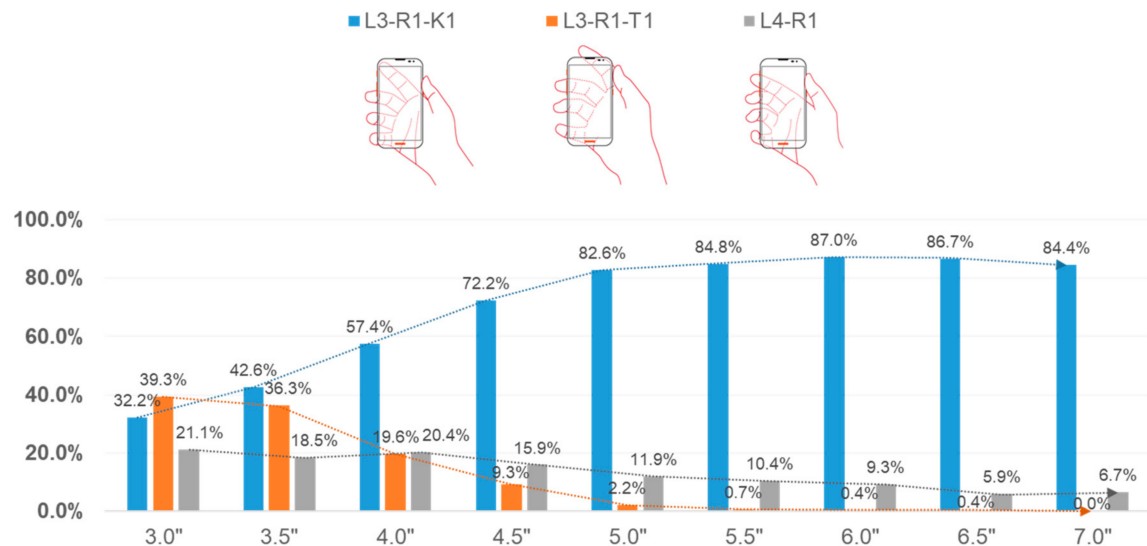

**Figure 5.** The frequency distribution of grip posture by smartphone size for hard key operations on a smartphone.

Lastly, the frequency distribution of grip posture varied significantly by hand width and hand length with a similar pattern as shown in Figure 6 ($\chi^2(4)$ = 75.3, $p$ < 0.001 for hand width and $\chi^2(4)$ = 103.4, $p$ < 0.001 for hand length). As hand width and hand length increased from small to large, the frequency of L3-R1-K1 decreased from 77.4% to 64.3% and from 77.2% to 64.0%, respectively. On the other hand, as hand width and hand length increased from small to large, the frequency of L4-R1 increased from 7.3% to 20.2% and from 5.6% to 21.9%, respectively. Lastly, as hand width increased from small to large, the frequency of L3-R1-T1 increased from 9.6% to 15.1% and then decreased to 11.4%; likewise, as hand length increased from small to large, the frequency of L3-R1-T1 increased from 10.7% to 14.1% and then decreased to 11.0%.

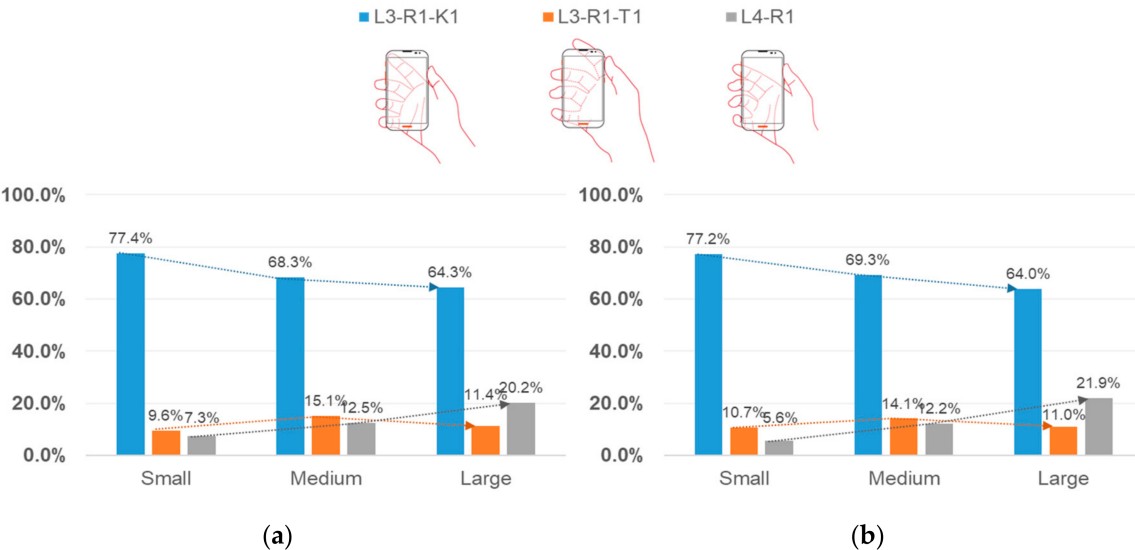

**Figure 6.** The frequency distribution of grip posture by hand size for hard key operations on a smartphone. (**a**) use frequency distribution of grip posture by hand width; (**b**) use frequency distribution of grip posture by hand length.

## 4. Discussion

The present study analyzed preferred grip postures by users with different hand sizes and their frequencies in one-handed operations of the smartphone hard keys (power and volume keys), which are needed to determine optimal locations of hard keys on a smartphone for efficiency and comfort. Properly designed hard key locations based on an analysis of user-preferred grip postures can enhance grip stability and reduce discomfort in the use of a smartphone [1,2]. Although one-handed smartphone operations cause more discomfort and usability problems than two-handed smartphone operations, users often prefer one-handed operations for efficiency and convenience reasons [17]. However, systematic studies on one-handed smartphone grip postures in hard key operations considering smartphone size and hand size have not been conducted. In the present study, one-handed smartphone grip postures were captured by camera and encoded by the number of fingers at each side of a smartphone. Dominant grip postures in hard key operations on smartphones were identified by analyzing the frequency distribution of grip posture, which can be used to determine the proper hard key locations of a smartphone with a particular size.

The smartphone grip posture analysis method employed in the present study based on the number of fingers at each side of a smartphone was found to be effective in describing various smartphone grip postures and identifying dominant grip postures in terms of frequency. Two web cameras vertically positioned apart enabled the hand in grip position on a smartphone to be efficiently captured. Ref. [7] classified smartphone grip postures into soft and firm grips by visual inspection, while the present study quantitatively classified grip postures by the number of fingers at each side of the phone. The proposed method can be applied to measurement and analysis of grip postures for operations of not only hard keys but also graphical user interfaces (GUI) on a touch screen and other smartphone physical user interfaces such as a fingerprint sensor.

The present study identified that nine grip postures are used for hard key operations, and out of those nine, three postures are dominant with a total frequency of 95.3% for smartphones of 3.0 to 7.0-inch screen size. The dominant grip postures include L3-R1-K1 (last three fingers at the left side, thumb at the right side, and index finger at the back of smartphone), L4-R1 (four fingers at the left side and thumb at the right side of smartphone), and L3-R1-T1 (last three fingers at the left side, thumb at the right side, and index finger at the top of smartphone). The dominant grip postures are similar in the aspect that the middle, ring, and little fingers are located at the left side and thumb at the right side of a smartphone, but are different in the aspect that the index finger is located at the back of a smartphone for L3-R1-K1, the left side for L4-R1, and the top for L3-R1-T1. The dominant grip postures enable the right hard keys to be easily operated by the thumb and the left hard keys by the index or middle finger while the device is being firmly held. In particular, L3-R1-K1 was found to be most dominant with a frequency of 70.0% because it enables users to switch fast between L4-R1 and L3-R1-T1 by moving the index finger to the left side and the top of a smartphone. Note that the subsequent three dominant grip postures, L2-R1-K1-B1, L3-R1-B1, and L2-R1-T1-B1 (total of frequencies = 4.7%), slightly differ from the three dominant postures in that the little finger is positioned to the bottom of a smartphone, instead of the left side, to support the base of the device more firmly.

The frequency distribution of grip posture by smartphone size identified that dominant grip postures with > 10% of frequency change by smartphone size: L3-R1-K1, L4-R1, and L3-R1-T1 for 3.0 to 4.0 inches of screen size, L3-R1-K1 and L4-R1 for 4.5 to 5.5 inches, and only L3-R1-K1 for 6.0 to 7.0 inches. The change in grip posture frequency distribution by smartphone size can be explained by the transition from holding small smartphones with a low height (e.g., 95 mm) by L3-R1-T1 or ones with a narrow width (e.g., 56 mm) by L4-R1 to holding large smartphones with a high height (e.g., 175 mm) or ones with a wide width (e.g., 93 mm) by L3-R1-K1, because users tend to move their index finger on the top or left-side for small smartphones and to the back for large smartphones for secure grip and support.

Next, the frequency distribution of grip posture by hand size identified an increase in L4-R1 by 12.9% and 16.3% and a decrease in L3-R1-K1 by 13.1% and 13.2% as hand width and hand length

increased from small to large, respectively. The change in grip posture frequency distribution by hand size can be explained by the observation that the index finger can be located more naturally at the left side of a smartphone as hand size increases because users with a large hand usually grasp a smartphone along a diagonal direction of their hand with a straight wrist posture while keeping the smartphone display vertical.

The findings of the present study need to be verified by users of a wider range of ages and having hand sizes beyond the hand size range of the present study and in use contexts other than standing. Users younger or older than their 20s and having smaller or larger hands than the hand size range of the participants in the present study need to be included to generalize the grip posture analysis results of the present study to the global user population because possible effects of age and hand size on grip posture beyond the ranges of age and hand size not explored in the present study. Next, the present study measured comfortable grip postures while users simulated smartphone tasks while standing in a laboratory environment; thus, grip postures of a smartphone in other use contexts such as sitting, lying, and walking need to be examined. Although the standing use context was selected in the present study for ease and efficiency in grip posture measurement, other use contexts need to be considered to examine if the distribution of grip posture changes depending on use context. For example, the grip posture of L3-R1-K1 preferred in the standing use context can be changed to L4-R1 in the walking use context to grasp a smartphone more firmly due to an increased likelihood of a drop and therefore damage to the device. Novel methods of grip posture measurement in natural and dynamic use contexts need to be developed to investigate user-preferred grip postures in various use contexts. Moreover, grip postures with real smartphones need to be investigated since mockups were used in the present study. Users can use different grip postures when holding the mockups made of white Acrylonitrile Butadiene Styrene (ABS) by 3D printer compared to holding real smartphones manufactured by molding and coated by various finishes of high quality. In addition, users can grasp in a safer way when using real smartphones to prevent dropping and damage.

Lastly, research on the determination of optimal locations of hard keys on a smartphone using the identified dominant grip postures is needed. The proper locations of hard keys can be determined by identifying preferred areas for hard key operations with users with different hand sizes while the identified dominant grip postures are used. For example, the preferred area of the left hard key for L4-R1 would vary by hand size and be located higher than that for L3-R1-K1. Thus, the optimal locations of hard keys on a smartphone of a particular size can be determined by considering hard key locations preferred by the user population with various hand sizes while the smartphone is held by the identified dominant grip postures. For example, the locations for hard keys can be determined by finding the comfortably reachable locations in the grip postures of L3-R1-K1, L4-R1, and L3-R1-T1 for users with various hand sizes. The ergonomically determined hard key locations can be evaluated in terms of time efficiency, accuracy, and comfort for validation [2].

## 5. Conclusions

The present study examined preferred grip postures on smartphone mock-ups in different screen sizes of 3.0″–7.0″ by users with different hand sizes in one-handed hard key operations while simulating major smartphone tasks while standing in a laboratory experiment. The grip postures were encoded by the locations (left side: L, right side: R, top: T, bottom: B, front: F, and back: K) on a smartphone and the number of fingers at each contact location. A total of nine grip postures were identified as those used for hard key operations, and out of those nine, three postures (L3-R1-K1: 70.0%, L4-R1: 13.3%, L3-R1-T1: 12.0%) were found to be dominant. Changes in grip posture frequency distribution by smartphone size and hand size were identified—the larger the smartphone size or hand size, the higher the frequency of L3-R1-K1. The grip posture frequency distribution by smartphone size would be of use to determine the optimal locations of hard keys on a smartphone of a particular size. The findings of the present study need to be verified further by users with hand sizes beyond the hand size range of the present study and use contexts other than standing.

**Author Contributions:** Conceptualization, Y.C. and H.Y.; methodology, Y.C., J.P., and H.Y.; software, J.P. and Y.C.; validation, Y.C., X.Y. and W.L.; formal analysis, Y.C. and W.L.; investigation, Y.C. and J.P.; data curation, Y.C. and J.P.; writing—original draft preparation, Y.C. and X.Y.; writing—review and editing, W.L. and H.Y.; visualization, Y.C.; supervision, H.Y.; project administration, H.Y.; funding acquisition, H.Y. All authors have read and agreed to the published version of the manuscript.

**Funding:** This research was jointly supported by the research programs (2017M3C1B6070526; 2018R1A2A2A 05023299) through the National Research Foundation of Korea (NRF) funded by the Ministry of Education, Science and Technology (MEST), those (No. 10063384; R0004840, 2017) of the Ministry of Trade, Industry, and Energy (MOTIE) under Industrial Technology Innovation Program, and the Biomedical Research Institute Fund, Chonbuk National University Hospital.

**Conflicts of Interest:** The authors declare no conflict of interest. The funders had no role in the design of the study; in the collection, analyses, or interpretation of data; in the writing of the manuscript, or in the decision to publish the results.

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
