# Peer review of "Effects of Smartphone Size and Hand Size on Grip Posture in One-Handed Hard Key Operations"

_applsci, doi:10.3390/app10238374_

Round 1

Reviewer 1 Report

The paper is well written and the addressed problem may be relevant, even though for a very specific subset of readers (not in general). However, I have several major problems with the manuscript, which prevent me from recommending its publication in its present form.

  1. The population addressed in the experiment encompasses young people (approx aged 20-29). Even though this population could be representative of the most extensive smartphone users, I am not sure that the way they grip and use smartphones is typical of other age ranges. Hence, if the authors intend to refer the results of their experiments to this age range only, they should clearly state this; otherwise, they should discuss and clarify this point.
  2. Another key issue is that I am not convinced that using mockups reflects the way one uses a true smarphone, especially for tasks such as web browsing. Are your experiments realistic of true smartphone using? Please provide information and possibly a validation of this issue.
  3. As for the experimental procedure, I understand that video is captured using two web cameras. However, no detail is provided on how these video sequences are processed. Do you address simple visual inspection? Is some detection algorithm is implemented? This is a major point, because the techical soundness of the manuscript cannot be evaluated in the absence of such information.
  4. Minor issues. Page 2 lines 74-75: Please provide definitions of t_{44} and f_{44,3920}. Page 3: Fig. 2 is cut out  and cannot be fully displayed.

Author Response

The paper is well written, and the addressed problem may be relevant, even though for a very specific subset of readers (not in general). However, I have several major problems with the manuscript, which prevent me from recommending its publication in its present form.

1. The population addressed in the experiment encompasses young people (approximately aged 20-29). Even though this population could be representative of the most extensive smartphone users, I am not sure that the way they grip and use smartphones is typical of other age ranges. Hence, if the authors intend to refer the results of their experiments to this age range only, they should clearly state this; otherwise, they should discuss and clarify this point.
=> The effect of age was not considered in this study. As recommended by the reviewer, the age issue of the participants has been addressed as a limitation of the study in the discussion section.

2. Another key issue is that I am not convinced that using mockups reflects the way one uses a true smartphone, especially for tasks such as web browsing. Are your experiments realistic of true smartphone using? Please provide information and possibly a validation of this issue.
=> As recommended by the reviewer, the limitation of use of the mockups in the study has been addressed in the discussion section. Furthermore, the following has been added to the Materials and Methods section to explain our efforts to overcome the limitation of use of the mockups: “A sheet of paper screen was glued on each mockup to provide a graphical user interface when simulating smartphone tasks.”

3. As for the experimental procedure, I understand that video is captured using two web cameras. However, no detail is provided on how these video sequences are processed. Do you address simple visual inspection? Is some detection algorithm is implemented? This is a major point, because the technical soundness of the manuscript cannot be evaluated in the absence of such information.
=> Details of video analysis sequence have been explained in the Materials and Methods section as follows: “Image frames of hard-key manipulation actions were extracted from the recorded video images. Then, each image was manually encoded by indicating the locations (left side: L, right side: R, top: T, bottom: B, front: F, and back: K) of the mockup and the number of fingers at each contact location.”

4. Minor issues. Page 2 lines 74-75: Please provide definitions of t_{44} and F_{44,3920}.

=> Statistics of t-test and F-test have been clearly indicated.

5. Page 3: Fig. 2 is cut out and cannot be fully displayed.
=> Fig. 2 has been corrected.

Reviewer 2 Report

The authors write about an issue that, although of utmost importance in what concerns with the design of mobile apps, has not, to my knowledge, been extensively covered in the literature. The paper is well written and the conclusions seem relevant. For this reason, I recommend its publication.

Author Response

The authors write about an issue that, although of utmost importance in what concerns with the design of mobile apps, has not, to my knowledge, been extensively covered in the literature. The paper is well written and the conclusions seem relevant. For this reason, I recommend its publication.

=> We appreciate your positive review our manuscript for publication.

Reviewer 3 Report

It is a good article with an interesting topic for the telephony industry that must consider the ergonomic positions of the hand in the manufacture of cell phone types.

I appreciate the experiment part which comprised the analyse of 2430 different grip posture.
However, I think the number of participants in the study is quite small considering the variability in the length and width of the hands (9 groups out of 45 participants).

I have some small adjustment I would like the authors to do.

In the introduction there are some citations which are not properly made, which make the article more difficult to read. Ex: [2] compared four one-handed grip posture..... instead of Lee et al. compared four.....and writing the citation at the end of the sentence. (lines 48,53,55,56)

In the participants field the authors should mention if the participants were also healthy in terms of neurological or visual impairments (if they asked or evaluate them).

I have a question regarding the preferred grip posture used. Were participants asked to perform the task from the safest grip posture (regarding the possibility of dropping the mockup) or from the most comfortable grip posture? Sometimes people tend to maintain a more secure posture on the phone, not always a comfortable posture.

In line 209 the authors mention about the subsequent three dominant grip postures referring to L2-R1-K1-B1, L3-R1-B1 and L2-R1-T1-B1 with a total frequency of 4.2%. I think that it is a mistake because in figure 4 the fourth preferred posture is L2-R1-K2 with a frequency of 2,4% and the sum of frequencies from L2-R1-K1-B1, L3-R1-B1 and L2-R1-T1-B1 is 2.1%. Please correct or explain if I didn`t understand.

It would be of interest to suggest what kind of hard key are suitable for the left side of the phone.

Author Response

It is a good article with an interesting topic for the telephony industry that must consider the ergonomic positions of the hand in the manufacture of cell phone types.

I appreciate the experiment part which comprised the analyse of 2430 different grip posture.
However, I think the number of participants in the study is quite small considering the variability in the length and width of the hands (9 groups out of 45 participants).
=> The representativeness for the participants (n = 45) for the Korean population in terms of hand length and hand width were checked by t-test for mean and F-test for variance. The limitations of the participant group in terms of hand size as well as age have been addressed in the Discussion section.

I have some small adjustment I would like the authors to do.

In the introduction there are some citations which are not properly made, which make the article more difficult to read. Ex: [2] compared four one-handed grip posture..... instead of Lee et al. compared four.....and writing the citation at the end of the sentence. (lines 48,53,55,56)
=> The citations have been corrected by following the reviewer’s suggestion.

In the participants field the authors should mention if the participants were also healthy in terms of neurological or visual impairments (if they asked or evaluate them).
=> The recruitment criterion of health condition for participants in the study has been added in the Materials and Methods section by following the reviewer’s comment.

I have a question regarding the preferred grip posture used. Were participants asked to perform the task from the safest grip posture (regarding the possibility of dropping the mockup) or from the most comfortable grip posture? Sometimes people tend to maintain a more secure posture on the phone, not always a comfortable posture.
=> We asked the participants to use comfortable grip postures rather than safe grip postures while simulating smartphone tasks with the mockups in standing. We agree that grip postures can vary from comfortable grip postures to safe grip postures depending on use context—for example, safe grip postures can be used in situations such as walking, running, and lying down. We have addressed this grip posture change issue depending on use context in the Discussion section.

In line 209 the authors mention about the subsequent three dominant grip postures referring to L2-R1-K1-B1, L3-R1-B1 and L2-R1-T1-B1 with a total frequency of 4.2%. I think that it is a mistake because in figure 4 the fourth preferred posture is L2-R1-K2 with a frequency of 2,4% and the sum of frequencies from L2-R1-K1-B1, L3-R1-B1 and L2-R1-T1-B1 is 2.1%. Please correct or explain if I didn`t understand.
=> The total frequency of 2.4% has been corrected to 4.7%.

It would be of interest to suggest what kind of hard key are suitable for the left side of the phone.
=> A recommendation for the locations for hard keys has been added in the Discussion section as follows: “For example, the locations for hard keys can be determined by finding the comfortably reachable locations in the grip postures of L3-R1-K1, L4-R1, and L3-R1-T1 for users with various hand sizes.”

Round 2

Reviewer 1 Report

The authors have taken into account my comments, even though mainly as reported limitations of the study itself, and without significant changes in the paper mainstream. However, as these limitations are honestly reported, I feel that the paper may be published it its present form.